# Unveiling the Actual Functions of Awns in Grasses: From Yield Potential to Quality Traits

**DOI:** 10.3390/ijms21207593

**Published:** 2020-10-14

**Authors:** Fabrice Ntakirutimana, Wengang Xie

**Affiliations:** State Key Laboratory of Grassland Agro-Ecosystems, Key Laboratory of Grassland Livestock Industry Innovation, Ministry of Agriculture, College of Pastoral Agriculture Science and Technology, Lanzhou University, Lanzhou 730020, China; ntakirutimana16@lzu.edu.cn

**Keywords:** awns, grass species, photosynthesis, molecular genetic factors, grain filling, grain yield, grain quality, grain shattering, biomass yield, forage quality

## Abstract

Awns, which are either bristles or hair-like outgrowths of lemmas in the florets, are one of the typical morphological characteristics of grass species. These stiff structures contribute to grain dispersal and burial and fend off animal predators. However, their phenotypic and genetic associations with traits deciding potential yield and quality are not fully understood. Awns appear to improve photosynthesis, provide assimilates for grain filling, thus contributing to the final grain yield, especially under temperature- and water-stress conditions. Long awns, however, represent a competing sink with developing kernels for photosynthates, which can reduce grain yield under favorable conditions. In addition, long awns can hamper postharvest handling, storage, and processing activities. Overall, little is known about the elusive role of awns, thus, this review summarizes what is known about the effect of awns on grain yield and biomass yield, grain nutritional value, and forage-quality attributes. The influence of awns on the agronomic performance of grasses seems to be associated with environmental and genetic factors and varies in different stages of plant development. The contribution of awns to yield traits and quality features previously documented in major cereal crops, such as rice, barley, and wheat, emphasizes that awns can be targeted for yield and quality improvement and may advance research aimed at identifying the phenotypic effects of morphological traits in grasses.

## 1. Introduction

Through evolutionary processes, grass species have evolved specific organs with the potential to improve their survival and performance in constantly changing environments. The formation of the awn at the tip end of the lemma is one of the prominent examples of such evolution [1]. Awns are essential, especially under natural conditions, for grain dispersal and deterring herbivores from ingesting grains [2,3]. These stiff structures are not only hygroscopically responsive, which facilitate self-planting but also maintain seed dormancy at maturity by avoiding pre-harvest sprouting [4]. This feature is unique and has not been achieved by any other monophyletic group of the plant kingdom. Awns have potential impacts on plant performance through their known functions during grain development, such as carbon accumulation and remobilization, and improved water-use efficiency [5,6,7]. How awns interact with yield and grain quality features determines how they sense stress signals and adapt to changing environments, and how these processes could translate into phenotypic variation are important biological questions. Exposing awns’ characteristics and their hidden interactions with kernel traits can have practical implications by providing a means of maintaining yield and quality in grasses. This will be achieved by examining direct and indirect influences of awns on individual components that underlie yield and quality, which will not only break barriers to understanding the potential roles of awns but also shed new light on the physiology of yield and quality traits of grasses. In this review we use primarily the physiological processes related to cereal crops, including rice (*Oryza sativa* L.), wheat (*Triticum aestivum* L.), barley (*Hordeum vulgare* L.), and rye (*Secale cereale* L.), and insights from some forage grasses to discuss and summarize the impacts of awns on specific grain traits. We suggest that the functional roles of awns are mainly associated with awn length and growing conditions (Figure 1) and that advances in both field and laboratory trials that combine both phenotypic and genomic approaches could provide a powerful window into understanding the actual roles of awns.

It is worth noting that this review uses the term ”grass” to refer any plant of the family Poaceae, ”cereal” to refer a type of grass grown for the edible grains, including major cereals (for example, wheat, rice, and barley) and minor cereals, such as rye. We also use the term” forage grasses” to refer those plants from the grass family cultivated mainly for cut fodder or grazed pastures.

## 2. Impact of Awns on Grain Yield and Plant Biomass

### 2.1. Awns and Their Contribution to Canopy Temperature

The capacity of awns to limit water loss persists beyond grain maturation. The increased transpiration by awns, particularly under water deficit, is mainly induced by the presence of stomata on the awn surface [8]. In different wheat cultivars, for example, awns have up to 50% of the total stomata in the spike and have more stomata than flag leaves [2]. Besides, awns of most barleys and wheats possess xeromorphic features, such as thick epidermis, cuticles, and high conductive tissues, providing water-use efficiency, especially under conditions of high temperature and water shortage [9,10]. As a result, awns increase spike transpiration in wheat and barley by up to 10 percent [5].

Although some studies have attempted to ascertain the role of awns in plant thermoregulation [6,11], how awns sense high-temperature signals and adapt to adverse conditions is not well understood. This is because reliable measures of the inflorescence temperature are hard to obtain. Consequently, the available information is mostly related to canopy temperature. Investigations of awned-awnless near-isogenic lines (NILs) of barley [12] and durum wheat [8] reported awned NILs to be cooler with a noticeable contribution to the canopy temperature reduction. Consistent with this observation, the research by Maydup et al. [6] in wheat used awn excision treatment and observed that plants that retained their awns were cooler in the morning and warmer in the afternoon compared with de-awned treatments. However, it is unclear if this thermoregulation potential of awns should be simply linked with awn morphology, or whether other factors, such as growing conditions and molecular genetic factors, could be involved. Besides, proteomic analyses in wheat showed that drought-responsive differentially accumulated proteins (DAPS) were up-regulated in awns, providing genetic evidence that awns are relatively resistant to drought and high temperature [13]. Determining if such DAPs are more up-regulated in awns than flag leaves and other photosynthetic organs will require further work. Therefore, the influence of awns on canopy temperature is meaningful since differences in canopy temperature could translate into various consequences. For instance, increased temperature should result in higher photo-respiration, compromising CO_2_ assimilation [14]. High temperatures could also hinder grain filling and possibly reduce grain weight. Cooler canopy temperature by transpiration efficiency could result in higher photosynthetic gas exchange [6]. Thus, higher grain yield could be achieved by optimizing awn length to provide a surface for cooling through transpiration efficiency.

### 2.2. The Photosynthetic Role of Awns and Their Possible Contribution to Grain Filling

In awned species, the amount of photosynthetic assimilation appears to directly affect grain filling [15,16,17]. Awns of several grasses, such as barley, wheat, and rye, are triangular-shaped in cross-section with chlorenchyma cells and provide a large photosynthetic surface (Figure 2), accounting for up to 60% of the total inflorescence surface area [18]. This explains why rice awns are not photosynthetically active since they lack chlorenchyma [19]. Because awns are located at the top of the canopy where light is abundant, they are suitable for light interception and CO_2_ uptake, so their photosynthetic role has been reported to be large and can account for as much as 90% of total spike photosynthesis in barley [20] and between 40–80% in wheat [5]. According to Evans et al. [21], wheat awns can double the net assimilation rate for grain filling under well-watered regimes, and can contribute between 34–43% when the moisture supply is limited. Olugbemi et al. [22] reported awns of wheat NILs contributed up to 12% to total canopy photosynthesis under moist conditions. Since the translocation pathway from the awn to the grain is short, awns can easily improve the photosynthetic capacity of the inflorescence [23]. A large photosynthetic role of awns was also observed by chloroplast analyses [16]. Chloroplasts are the sites of photosynthesis and their development appears to be linked with variations in photosynthetic activity [24]. In flag leaves, chloroplasts develop in the early stages of floral emergence and possess well-structured thylakoids. However, near grain-development stages, the leaves senesce quickly, triggering the degradation of grana thylakoids and, thus, producing small amounts of starch. In contrast, the chloroplasts of awns develop near the grain-filling stage and remain functionally active until grain maturity with a large number of grana stacks, indicating that awns contribute more to carbon assimilation than flag leaves during the last stages of grain filling [16].

Studies investigating the photosynthetic oxygen evolution, the amount of oxygen produced by the photosystem II (PSII) protein-cofactor complex embedded in the thylakoids in wheat, demonstrated that the rate of oxygen evolution was higher in the flag leaves than in the awns before the dough stage and this difference was nearly twice as high at the milk stage [25]. Intriguingly, the rate in the awns increased considerably during grain development, whereas that of the flag leaves decreased gradually. Assays of phosphoenolpyruvate carboxylase (PEPCase) activity revealed the photosynthetic role of awns [14]. PEPCase is the enzyme responsible for the initial carbon fixation in C_4_ and crassulacean acid metabolism (CAM) plants, and for replenishing tricarboxylic acid cycle intermediates and for the operation of stomatal guard cells in C_3_ plants [26]. Wheat awns showed substantially higher PEPCase activity than flag leaves, particularly at the late stages of grain development [16], but the ability of awns to assimilate CO_2_ by C_4_ photosynthetic pathway remains to be determined [27]. Another study used the stable isotope discrimination (Δ^13^C) to investigate traits contributing to grain filling in hexaploid wheat and observed awns had higher Δ^13^C than flag leaves under well-watered condition [28]. According to Merah and Monneveux [29], the carbon isotope discrimination of wheat awns was more associated with grain yield than that of flag leaves under rainfed conditions. Furthermore, a recent genomic scan in barley has found that among 11 genes known to participate in the Calvin cycle, nine were preferentially expressed in the awn [20]. The authors also observed that genes for the biosynthesis of chlorophyll and carotenoids, and reactive oxygen species scavenging were highly expressed in the awn. In wheat, proteins associated with photosynthesis, such as the chlorophyll-binding protein CBP8, the photosystem II proteins PsbO and PsbP, and the cytochrome b6f (cytb6f) complex, were highly expressed in awns, and downregulation of these proteins was associated with decreased starch biosynthesis and grain yield, especially under water deficit [13]. Thus, these observations serve as a foundation to understanding the photosynthetic nature of awns, especially in barley and wheat, yet they also reflect a need to unravel the photosynthetic potential of awns in other grass species.

### 2.3. Impact of Awns on Grain Yield and Components

Grain yield in grass species is the product of grain yield components, molecular genetic factors, and growing conditions and their interaction [30]. For the most part, grain yield is decided by grain number and grain weight [31], with increased grain weight being linked with assimilate availability and distribution [32,33]. Studies investigating the phenotypic evolution of grasses suggest that awns significantly improve grain weight, given their contribution to assimilate production and partitioning [6,34,35,36,37,38]. This is particularly true for wheat [39,40] and barley [41,42], where removing the awns significantly reduced grain weight and grain yield. Besides, according to Grundbacher [43], awns could improve grain yield in rye under standard growing conditions. This is in contrast with several other studies where long awns required a portion of assimilates for development, which decreased the assimilate accumulated in the kernel resulting in yield loss under irrigated conditions [44,45]. Other studies observed little difference in grain yield of awned and awnless wheat NILs under moist conditions [46,47]. In several cultivars of rice, long awns showed a neutral impact on grain yield under water-limited conditions, and they decreased the total harvestable grain yield under irrigated conditions [48]. The negative impact of long awns was also observed in barley under terminal drought [49]. However, Li et al. [16] concluded that the impact of awns on grain yield was uncertain, especially for short-awned species, since the grass inflorescence comprises other photosynthetically active organs, such as palea, lemma, glumes, and seed pericarp. Besides, in some species, the contribution of awns during grain filling could be compensated or possibly exceeded by assimilates used during awn development [30,50]. Based on these results, we hypothesize the phenotypic role of awns varies with the species, environment, and awn length (Table 1).

That the activity of awns is affected by environmental factors is supported by data from several studies focused on the functional roles of awns, with the effect of moisture frequently overlooked. It is worth noting that moisture may not always be the deciding factor, as the environments could also vary in temperature, radiation, and soil fertility [44]. Our study in Siberian wildrye (*Elymus sibiricus* L.) demonstrated that the contribution of awns to grain yield and yield components, including grains per inflorescence, thousand-kernel weight, and grain size, varied across latitudes, longitudes, and altitudes [35]. This indicated not only a complex interaction between environment and awn features but also pointed to multi-lateral impacts of this interaction. Of note, many of the reported roles of awns are only based on wheat and barley and the information on other grass species is not widely available. For this reason, it is important to test the pleiotropic effects of awns in additional cultivars and/or various awned and awnless NILs from a range of diverse grass species under different conditions (years, locations, and management practices).

While most pathways involved in the development of awns were characterized molecularly mainly in rice [52], the genetic basis of how awns affect yield-related traits and how they influence grain yield is not well understood. A number of approaches including whole-genome sequencing, examination of candidate genes, and quantitative trait locus studies, have provided an overview on genes involved in awn development [53,54]. For example, several genes involved in the development of awns in rice, including *An-1* [55], *An-2* [56], *Laba1* [1], *Gad1* [19], and *RAE3* [15], were characterized during the past decade. The current understanding is that many of these genes interact with grain number per panicle and kernel size and weight, but the debate regarding their effects on final grain yield is ongoing [3,57]. Analyses of *Gad1*, for example, suggested a neutral effect of this gene on grain yield, which was mostly driven by a balance between grain size and grains per panicle. On the other hand, molecular analyses of *An-1* [55], *An-2* [56], and *Laba1* [1] indicated that these genes were associated with decreased total grain yield compared to their mutant alleles. In barley, *Lks1* and *Lks2* genes, associated with awn development, caused differences in floral forms [18], but little is known about their molecular linkage with yield-related traits. Liller et al. [58] identified *AL7.1*, the major quantitative trait locus (QTL) associated with long awns on barley chromosome arm 7HL, and with exception of little effect on grain width, no other yield-related traits were affected by this QTL. In wheat, the molecular basis of awn development is controlled by three dominant alleles identified as *Hooded1 (Hd)*, *Tipped1 (B1)*, and *Tipped2 (B2)*, mapped on, respectively, chromosome arms 4AS, 5AL, and 6BL [2]. The identification of these loci offers intriguing insights into the regulatory network underlying awn suppression in wheat [54] although information regarding their impact on grain yield is lacking. In summary, the available information on the regulatory network underlying the biological impact of awns opens up a suite of interesting new questions, which once resolved, should provide essential breakthroughs in our understanding of genetic and developmental inputs that determine grain yield.

### 2.4. Role of Awns on Biomass Yield

Biomass yield has been the focus in the commercial release of new forage-grass cultivars. A promising avenue for improving biomass yield is to balance the allocation of assimilates between growing florets and traits that determine fodder yield. A number of phenotypic traits, such as plant height, stem girth, and number of tillers, have a direct influence on biomass yield [59]. Until recently, no direct relationship was documented between awns and these traits. Previous analyses showed that inflorescence dry weights and the harvest index of awned wheat genotypes were greater than those of awnless ones [6,45], but it is not clear whether this difference is linked with awn length. Pampana et al. [60] documented a negative correlation between grain yield and total biomass in modern wheat cultivars. Given that awns can increase available assimilates for growing grains, we hypothesize that awns contribute to grain yield rather than to biomass yield. This hypothesis is difficult to evaluate because, in some species, there might be inconsistent differences in dry matter partitioning to the developing grains between short-awned and long-awned genotypes [8]. Besides, awned and awnless genotypes from diverse genetic backgrounds may have different biomass yield [45]. Is biomass yield simply associated with genotype and growing conditions, or could awns indirectly interact with traits determining biomass yield? More generally, do awns interact with genetic pathways underlying grain yield rather than biomass yield? Molecular genetic approaches elucidating whether awns interact with the gene-regulatory networks underlying biomass yield, coupled with carefully managed field experiments contrasting biomass yields of awned and awnless genotypes, could thus contribute to answering these questions.

### 2.5. The Impact of Awns on Grain Shattering

Grass cultivars have been bred primarily for grain yield and biomass production rather than high grain retention although grain shattering has severely hampered harvested yield in many domesticated grass species [61]. Grain shattering susceptibility is measured by the way mature grains are retained or detached from the pedicel at the abscission layers [62] and is generally induced by the progressive deterioration of the abscission layer, its middle lamellae, and cell walls [63]. In some cereals, such as rice and millet, grain shattering is triggered by the rachilla detaching just above the glumes. In barley and wheat, grain shattering is mainly caused by the rachis detaching just below the glumes. In some grass species, such as *Agropyron* spp. and cocksfoot (*Dactylis glomerata* L.), this breakage may occur at both positions [64].

Very few studies have evaluated the relationship between grain shattering and floral morphological structures. The effect of awns on grain shattering is poorly understood even though some studies have revealed awns might be linked to grain shattering mechanistically since long awns improve kernel weight, raise glume compactness, and increase the rachilla fragility by reducing pedicel-breaking tensile strength (BTS), thus, causing grain shattering [61,65]. Grain shattering is a highly coordinated process that appears to be linked with changes in cell structure, metabolism, and gene expression [66,67]. A number of studies have shown grain shattering relies on the degradation of abscission layers just at the base of the grain, while grain retention is caused by lack of abscission layers [68]. Abscission occurs as a result of cleavage of cell wall components by a rapid increase in cellulase and polygalacturonase activity [69]. This process seems to be governed by several hormones, including ethylene acting as a promoter, abscisic acid as a modulator, and auxin as an inhibitor [70]. How awns interact with these hormones and other abscission regulators is not well understood and remains a subject of intense investigation.

Abscission is a complex developmental process that involves several genetic factors. A genomic scan in rice detected the awn-length single nucleotide polymorphism (SNP), sf0136352825, nearby 95 kb away from the grain shattering gene *qSH1* on chromosome 1 [71]. Another study revealed the association between awn development gene (*An1*) and grain-shattering-related QTL (*sh4-1)* on chromosome 4 in an advanced backcross between *O. sativa* cultivar Jefferson and an accession of *O. rufipogon* (IRGC 105491), a wild ancestral species of rice [65]. However, these results are not strictly conclusive since cultivated rice typically has short or non-existent awns, although long-awn variants exist, and grain shattering is not a common trait in cultivated rice. Furthermore, grain dispersal by awns has been suppressed in some wheat and barley cultivars by insertion of the grain retention trait, which is governed by various regulators, including *naked caryopsis* (*nud*) and *thresh-1* genes in barley or *Tenacious glumes* (*Tg*) locus and the Q allele in wheat [72]. These findings emphasize the undocumented role of awns in grain shattering of some cereals and reinforce the need for additional research in other grass species to learn whether and how awns are linked with grain shattering.

## 3. Impact of Awns on Grain and Forage Quality Traits

### 3.1. Grain Quality Traits as Affected by Awns

The earlier concept of kernel quality in cereals has been based on germination behavior rather than nutritional values of these grass species. Improved nutritional composition of grains is an emerging target in plant breeding, particularly, as consumers express increasing concern about the poor grain quality of some modern cereal cultivars, such as rice and wheat [73,74]. Quality aspects related to awn characteristics have been of interest mainly during domestication where short-awned or awnless species have been preferred because they were easier to handle and process [18]. As indicated by several studies, the increase in nutritional values, such as carbohydrates and proteins, is coupled with an increase in the size of the endosperm. Grains with a larger endosperm provide higher carbohydrates but fewer proteins than those with larger embryos [75].

Awns of some species can improve grain size with a large endosperm, and therefore have an impact on grain commercial value, whereas awns of other species have been linked with reductions in grain protein content [45,76]. For example, the research of Fasah et al. [77] showed that an accession of *O. rufipogon* (IRGC 105491), an awned wild relative of rice, had about 25.0% amylose, 8.0% protein, and 2.2% fat, whereas an awnless rice cultivar MR219 (*O. sativa* subsp. *indica*) contained about 21.0% amylose, 7.9% protein, and 2.0% fat. In common wheat, a cross between Glenlea, an awnletted cultivar, and Era, an awned cultivar, showed awnletted lines contained about 146.33 g kg^−1^ of average grain protein concentration, which was slightly reduced to 144.66 g kg^−1^ in awned lines [51]. By contrast, a genetic analysis of Mendos, an awnless wheat cultivar, and an awned bread wheat, W21MMT70, revealed a stronger association between increased flour protein concentration and the presence of awns with a phenotypic variance ranging from 7 to 14% [78]. Furthermore, in some rice species, long awns were associated with high concentrations of harmful heavy metals, such as nickel, arsenic, lead, cadmium, and mercury [79]. These observations indicate differences in nutritional values of awned and awnless species, but whether or how these variations are directly linked with awn features has not been established. Finally, the influence that awns have on grain nutritional values varies across genotypes and the environments tested, irrespective of the fact that awns could increase test weight and reduce grain screenings to enhance market value [45]. Unveiling the impact of awns on various grain quality traits is necessary for the successful release of desirable commercial grass cultivars.

### 3.2. Impact of Awns on Forage Quality Traits

Forage quality refers to the potential of a given forage grass or legume to meet the palatability and nutritional needs of an animal. Although maturity, growing conditions, and genotype are principal factors affecting forage quality, several studies have found ample evidence that morphological features of the grass inflorescence can affect forage quality, especially at maturity [80,81,82,83]. Several studies have revealed that awns had a negative impact on forage quality [84,85] because, in contrast to short-awned and awnless species, long-awned species reduced total digestible nutrients, crude protein, and palatability, and might cause irritation and soreness to livestock [85,86]. Considering that awn length is extremely variable and can potentially be modified across grass species and that forage quality is affected by diverse abiotic and biotic factors, additional studies are needed to clarify the impact of awns on forage quality and document that awn characteristics can preserve both biomass yield and quality.

## 4. Conclusions and Future Perspectives

Awns serve several functions during grain development, most of which have evolved to improve carbohydrate production. Although awns cannot fully replace flag leaves as the main source of photosynthates, their contribution can be large, with a range of environmental conditions affecting canopy photosynthesis, including water shortage, high air temperatures, leaf diseases, and other factors resulting in premature leaf senescence. Awns have undergone phenotypic modification since domestication, thus determining the agronomic impact of awns remains an important but challenging goal for future studies in grasses. Currently, relatively little is known about the agronomic importance of awns because the phenotypes are part of dynamic developmental processes, which are governed by genetic and environmental cues and their interactions. Future studies should carefully assess the nature of these interactions and target efforts to introgress new elite phenotypes, adapting to local climate and end-use. Taking all these points into consideration, it is worth noting that assessments from variable genetic backgrounds, a wide range of environments, and numerous traits will aid in understanding the function of awns. Finally, we hope the research discussed here will inspire future studies not only to bring together analytical advances and the conceptual framework to enhance our understanding of awn functions but also to identify pathways to maximize yield and quality traits under normal growing conditions.

## Figures and Tables

**Figure 1 ijms-21-07593-f001:**
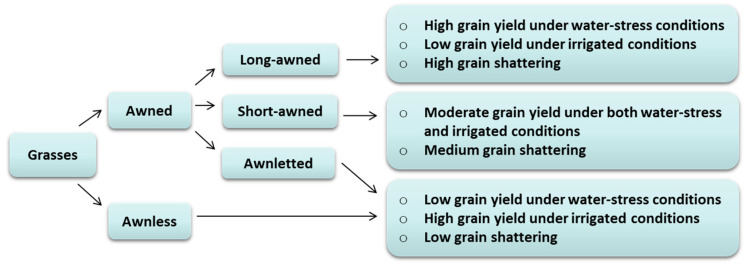
A schematic summary of awn length variation and its phenotypic roles. Morphologically, the awn is the extension of the lemma, which varies from several inches of length (as in long-awned) to total absence (as in awnless). Awnletted species have very short awns whilst awnless genotypes usually do not show awn development [8]. This classification is inconsistent and varies among species. The debate of how long awn trait has been lost in modern species of grasses compared to their wild relatives remains ongoing [1], although it has been hypothesized that long awn trait was selected against by ancient famers during domestication [2].

**Figure 2 ijms-21-07593-f002:**
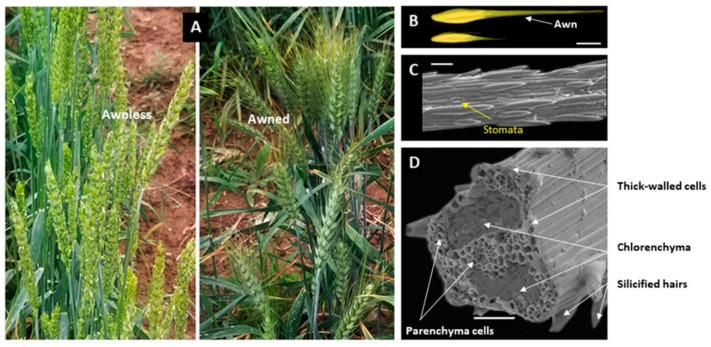
Structural characteristrics of the wheat awn. (**A**) Shows the spikes of awnless wheat (left) and awned wheat (right). (**B**) Shows the grains of awned wheat (top) and awnless wheat (down), bar = 1 cm; (**C**) shows the surface of the wheat awn under a scanning electron microscope, bar = 50 µm. (**D**) Shows a scanning electron micrograph of wheat awn cross-section, bar = 50 µm; the section of 2.5 µm was taken at 1 cm from the base of the awn; different tissues are illustrated by arrows.

**Table 1 ijms-21-07593-t001:** Relationships between awns, yield components, grain yield, and biomass yield in barley and wheat across different growing conditions.

Species	Growing Conditions	Genotypes Tested	Grains Per Inflorescence	Kernel Weight (mg)	Grain Yield Per Plant (g)	Biomass Yield (t ha^−1^)	References
Bread wheat (*Triticum aestivum* L.)	Dry land	Awned	36.90 ^a^	31.20 ^c^	1.13	nd	Teich [44]
		Awnless	37.60 ^a^	28.60 ^c^	1.03	nd	
	Irrigated	Awned	65.40 ^a^	31.40 ^c^	2.10	nd	
		Awnless	82.40 ^a^	30.60 ^c^	2.52	nd	
Bread wheat (*Triticum aestivum* L.)	Dry land	Awned	42.60	48.20	7.36 ^d^	nd	Duwayri [34]
		De-awned	36.10	45.80	6.19 ^d^	nd	
Bread wheat (*Triticum aestivum* L.)	Rainfed	Awned	Nd ^b^	34.13	305.00 ^e^	nd	Knott [51]
		Awnletted	Nd	32.50	291.33 ^e^	nd	
Bread wheat (*Triticum aestivum* L.)	Rainfed	Awned	61.95	40.96 ^c^	23.61	nd	Khaliq et al. [40]
		De-awned	59.45	39.12 ^c^	21.30	nd	
Bread wheat (*Triticum aestivum* L.)	Rainfed	Awned	23.30	39.00	2.84 ^d^	7.90	Rebetzke et al. [45]
		Awnletted	24.30	37.10	2.74 ^d^	7.59	
	Irrigated	Awned	25.90	40.80	6.27 ^d^	16.50	
		Awnletted	27.40	38.90	6.32 ^d^	16.20	
Durum wheat (*Triticum turgidum* L. var.*durum*)	Irrigated	Awned	45.54	54.60	238.00 ^f^	nd	Chhabra and Seth [39]
		De-awned	45.13	47.50	205.90 ^f^	nd	
Durum wheat (*Triticum turgidum* L. var. *durum*)	Rainfed	Awned	32.30	35.10	104.70 ^e^	nd	Motzo and Giunta [8]
	Rainfed	De-awned	31.20	34.50	94.10 ^e^	nd	
Barley (*Hordeum vulgare* L.)	Rainfed	Awned	42.40	46.90 ^c^	Nd	nd	Paluska [41]
		De-awned	44.70	37.80 ^c^	nd	nd	
Barley (*Hordeum vulgare* L.)	Irrigated	Awned	19.57	46.68	2.47	7.20	Hosseini et al. [49]
	Terminal drought	Awned	17.78	39.03	1.74	5.67	

^a^ grains per plant; ^b^ nd, not determined; ^c^ 1000 kernel weight (g); ^d^ In t ha^−1^; ^e^ In g m^−2^; ^f^ grain yield of 100 spikes (g). Note: Values within a column in the table that are not followed by small letters are in the unit provided in the column header. Values within columns in the table that are followed by the same small letter are in the unit of the corresponding small letter in footnotes.

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
