# Peer review of "Unveiling the Actual Functions of Awns in Grasses: From Yield Potential to Quality Traits"

_ijms, 2020, doi:10.3390/ijms21207593_

Round 1

Reviewer 1 Report

This review discusses current information on the influence of awns on agronomic traits in grasses and cereals. It is well written and readable and provides a useful summary, although the main conclusion seems to be that little is known about awn function and much more work is required. In this case, I believe that the title is misplaced as actual awn function is still uncertain with considerable contradictory data in the literature. However, in cereals, particularly wheat, there is a fairly clear correlation between the presence of awns and the regional climate, with awned varieties grown in hot, dry regions, while awnless varieties are preferred in cooler, wetter climates. This might have merited more discussion.

I have some minor comments.

A very recent publication (Sanchez-Bragado et al. Field Crops Research DOI: 10.1016/j.fcr.2020.107827) on the relevance of awns for yield in wheat, which I assume appeared after submission, should be discussed as it is highly relevant to the topic.

Line 35-37: what is the evidence that awn formation appeared by convergent evolution? Isn’t it more likely to have evolved in an ancestor of the Poaceae? It has been lost in some crop species during domestication.

Line 78: exacerbating is not the appropriate term here. Compromising may be better.

Line 94: spelling of photosynthesis.

Line 119: what is meant by overexpressed, which indicates unnaturally high expression? Highly expressed would be more appropriate.

Line 196: …compared with their mutant alleles?

Line 221: greater in short-awned genotypes than in long-awned ones?

Line 242: is it possible that long awns provide more leverage (moment) to promote grain shatter?

Line 247: it should be cellulase and polygalacturonase activity.

Line 250: perhaps provide some references to support the statement that the topic is under intense investigation.

Line 282: some explanation of the term awnletted is needed for readers not directly in the field.

Author Response

Reviewer 3:

This review discusses current information on the influence of awns on agronomic traits in grasses and cereals. It is well written and readable and provides a useful summary, although the main conclusion seems to be that little is known about awn function and much more work is required. In this case, I believe that the title is misplaced as actual awn function is still uncertain with considerable contradictory data in the literature. However, in cereals, particularly wheat, there is a fairly clear correlation between the presence of awns and the regional climate, with awned varieties grown in hot, dry regions, while awnless varieties are preferred in cooler, wetter climates. This might have merited more discussion.

Response: Thank for the time and efforts you have put into reviewing our ms. All your comments and suggestions are important to improving our ms. As you mentioned, there is clear relationship between awns and growing conditions. Thus, we have added information about this. Please check lines 52-54. We didn’t add much information because we didn’t want to duplicate the information from our previously published article (https://doi.org/10.3390/plants8120561).

Some minor comments:

A very recent publication (Sanchez-Bragado et al. Field Crops Research DOI: 10.1016/j.fcr.2020.107827) on the relevance of awns for yield in wheat, which I assume appeared after submission, should be discussed as it is highly relevant to the topic.

Response: We have added this reference as you suggested, please check reference no.30

Line 35-37: what is the evidence that awn formation appeared by convergent evolution? Isn’t it more likely to have evolved in an ancestor of the Poaceae? It has been lost in some crop species during domestication.

Reference: We agree with you that there is not enough information to prove if awns evolved through convergent evolution. Thus, we have revised this statement and prove more clear information. Please check line 35

Line 78: exacerbating is not the appropriate term here. Compromising may be better.

Response: We have corrected the statement as you suggested. Please check line 91.

Line 94: spelling of photosynthesis.

Response: We have corrected the term as you suggested. Please check the line 107.

Line 119: what is meant by overexpressed, which indicates unnaturally high expression? Highly expressed would be more appropriate.

Response: We have changed the term as you suggested. Please check line 138.

Line 196: …compared with their mutant alleles?

Response: We have corrected the term. Please check line 215.

Line 221: greater in short-awned genotypes than in long-awned ones?

Response: We have corrected the statement for more clarification. Please check lines 239-240.

Line 242: is it possible that long awns provide more leverage (moment) to promote grain shatter?

Response: From the previously published works, awns increase grain weight and thus reduces breaking tensile strength, which may cause the grain to easily shatter. This could also be connected with the ability of awns to influence seed dispersal. But, as you mentioned, this could be inconsistent in some species like rice, where grain shattering trait was lost in many, if not all, cultivated species. We have mentioned this statement in the ms, please check lines 276-277.

Line 247: it should be cellulase and polygalacturonase activity.

Response : We have corrected the terms as you mentioned, please check line 267

Line 250: perhaps provide some references to support the statement that the topic is under intense investigation.

Response: As you suggested, we would have added the reference to support the statement, but we could not find any reference related. Thus, we have stated that further works are needed.

Line 282: some explanation of the term awnletted is needed for readers not directly in the field.

Response:  As you suggested, we have explained the term. Please check lines 62-63.

We thank you again for valuable comments and suggestions.

Reviewer 2 Report

The authors describe a comprehensive review of the impacts of awns in a host of traits across different plant species. The manuscript is very well written, clearly presented and includes a nice summary of quantitative data from different field trials of impacts on yield traits. The paper provides a really detailed overview of current knowledge of awns on both quality and yield traits and indicates areas for future attention that will help us learn more about their impacts. 

I don't really have any suggestions for improvement, the bibliography appears complete to me and the authors are upfront when describing the limits of our knowledge. One area I might have like to read more from was in the genetic pathways that are regulating awn development. Given that several genes have been cloned, particularly in rice, it might have been good to describe the types of functions these genes have, whether there is an indication of them working in a common regulatory pathway and how this information might be utilized for targeted breeding and/or engineering awn physiology.  

Author Response

The authors describe a comprehensive review of the impacts of awns in a host of traits across different plant species. The manuscript is very well written, clearly presented and includes a nice summary of quantitative data from different field trials of impacts on yield traits. The paper provides a really detailed overview of current knowledge of awns on both quality and yield traits and indicates areas for future attention that will help us learn more about their impacts.

Response: Thank you for the time and efforts you have put into evaluating our ms. Your comments are all valuable and essential for the improvement of the ms and guidance for our further research projects.

I don't really have any suggestions for improvement, the bibliography appears complete to me and the authors are upfront when describing the limits of our knowledge. One area I might have like to read more from was in the genetic pathways that are regulating awn development. Given that several genes have been cloned, particularly in rice, it might have been good to describe the types of functions these genes have, whether there is an indication of them working in a common regulatory pathway and how this information might be utilized for targeted breeding and/or engineering awn physiology. 

Response: Thank you for your comments and suggestions. It is true that the ms didn’t talk that much about the genetic pathways that regulate awn development, though we mentioned some phenotypic role of some awn development genes and QTLs (Please check lines 201 to 222). The reason is that we have already published another comprehensive review about morphological and genetic factors associated with awn development in grasses (https://doi.org/10.3390/genes10080573) and we didn’t want to duplicate the information or mislead the readers of the ms.

We thank you again for the comments and suggestions.

Reviewer 3 Report

The article has been well written and is quite interesting. But what lacks in the mentioned text is the clear description of the major goal of your article.

Also, there should be a clear distinction between the use of nomenclature relating to grasses as such and to cereals. Of course, I know that cereals are also grasses, but it would be appropriate to make such a distinction in this type of work.

Considering the introduction, some more anatomic details about awns have to be given. A schematic graph presenting different types of awns or photographs may substantially increase the perception of your article.

Also, the aim of the publication was not clearly specified. What was your major goal, what did you want to describe or explain? Was there any hypothesis?

Below a few more details:

Line # 12: please, delete „A single paragraph of about 200 words maximum”.

Line # 43: describe what you do mean ‘quality’?  

Line  # 49: It is not common to use the term “cereal grasses”. It is better to use “cereals”…

Line # 78: space missing between ‘CO2 assimilation’

Line # 128 – 129. You mention grain yields but the cited reference deals with perennial ryegrass. Therefore, you should change ‘Grain yield’ into ‘Seed yield” or find much appropriate reference.

Line # 217: space missing between ‘negative correlation’

Author Response

The article has been well written and is quite interesting. But what lacks in the mentioned text is the clear description of the major goal of your article

Response: Thank you for the time and efforts you have put into assessing our ms. Regarding the major goals of the manuscript, we have added the information stating the goals of the article, as you suggested. Please check lines 49-54.

Also, there should be a clear distinction between the use of nomenclature relating to grasses as such and to cereals. Of course, I know that cereals are also grasses, but it would be appropriate to make such a distinction in this type of work.

Response: We thank you for the comment and suggestion. As you suggested, we have added information regarding the specific terms associated with grasses used in the ms. Please check lines 55-58.

Considering the introduction, some more anatomic details about awns have to be given. A schematic graph presenting different types of awns or photographs may substantially increase the perception of your article.

Response: We have added a schematic representation showing types of awns based on awn length (Please check figure 1). We have only focused on this because it was discussed throughout the ms and because in our previously published paper we have talk deeply about different awn types. (https://doi.org/10.3390/genes10080573). Besides, we have also added some photographs showing awns of wheat and their physical features (Please check figure 2).

Few more details:

Line # 12: please, delete „A single paragraph of about 200 words maximum”.

Response: We have deleted the paragraph as you suggested.

Line # 43: describe what you do mean ‘quality’? 

Response: We have explained the term as you suggested, please check line 43

Line  # 49: It is not common to use the term “cereal grasses”. It is better to use “cereals”…

Response: We have corrected the term. Please check line 49

Line # 78: space missing between ‘CO2 assimilation’

Response: We have inserted the spacing, please check the line 91

Line # 128 – 129. You mention grain yields but the cited reference deals with perennial ryegrass. Therefore, you should change ‘Grain yield’ into ‘Seed yield” or find much appropriate reference.

Response: We have changed the reference as you mentioned. Please check reference no. 30

Line # 217: space missing between ‘negative correlation

Response: We have inserted the spacing, please check line 236.

Thank you again for valuable comments and suggestions.